# Plant-expressed virus-like particles reveal the intricate maturation process of a eukaryotic virus

Roger Castells-Graells [1,5], Jonas R. S. Ribeiro[2], Tatiana Domitrovic[2], Emma L. Hesketh [3], Charlotte A. Scarff [3], John E. Johnson [4], Neil A. Ranson [3], David M. Lawson [1] & George P. Lomonossoff [1✉]

Many virus capsids undergo exquisitely choreographed maturation processes in their host cells to produce infectious virions, and these remain poorly understood. As a tool for studying virus maturation, we transiently expressed the capsid protein of the insect virus *Nudaurelia capensis* omega virus (NωV) in *Nicotiana benthamiana* and were able to purify both immature procapsids and mature capsids from infiltrated leaves by varying the expression time. Cryo-EM analysis of the plant-produced procapsids and mature capsids to 6.6 Å and 2.7 Å resolution, respectively, reveals that in addition to large scale rigid body motions, internal regions of the subunits are extensively remodelled during maturation, creating the active site required for autocatalytic cleavage and infectivity. The mature particles are biologically active in terms of their ability to lyse membranes and have a structure that is essentially identical to authentic virus. The ability to faithfully recapitulate and visualize a complex maturation process in plants, including the autocatalytic cleavage of the capsid protein, has revealed a ~30 Å translation-rotation of the subunits during maturation as well as conformational rearrangements in the N and C-terminal helical regions of each subunit.

[1] Department of Biological Chemistry, John Innes Centre, Colney, UK. [2] Virology Department, Instituto de Microbiologia Paulo de Goes, Universidade Federal do Rio de Janeiro, Rio de Janeiro, Brazil. [3] Astbury Centre for Structural Molecular Biology, School of Molecular & Cellular Biology, Faculty of Biological Sciences, University of Leeds, Leeds, UK. [4] Department of Integrative Structural and Computational Biology, The Scripps Research Institute, La Jolla, CA, USA. [5] Present address: Department of Chemistry and Biochemistry, University of California, Los Angeles, CA, USA. ✉email: george.lomonossoff@jic.ac.uk

Maturation is a critical part of the replication cycle of all animal and bacterial viruses studied to date and is required for the production of infectious virions. The outlines of maturation have been determined for retroviruses[1], flaviviruses[2], herpesvirus[3] and a variety of other viruses, but detailed mechanistic studies have not been reported. The maturation of particles of the insect virus, *Nudaurelia capensis omega virus* (NωV), has been studied extensively in vitro. As a result, the virus is now an exemplar for maturation processes[4,5]. NωV belongs to the *Alphatetraviridae*, a family of insect viruses with non-enveloped $T = 4$ capsids that infect a single order of insects, the Lepidoptera[6,7]. NωV has a single-stranded positive-sense, bipartite RNA genome consisting of RNA-1 (~5.3 kb), that encodes the RNA-dependent RNA polymerase, and RNA-2 (~2.5 kb), that encodes the 70 kDa α capsid protein[8]. When the coat protein of NωV is expressed in insect cells, it assembles into stable intermediate virus-like particles (VLPs) (known as the procapsid), 48 nm in diameter, that are porous at neutral pH (pH 7.6)[9]. These particles undergo a maturation process when they are exposed to acidic conditions (pH 5.0) in vitro. This involves large-scale subunit reorganisation and an autocatalytic cleavage of the α capsid protein between residues Asn570 and Phe571 to give two polypeptides, β and γ, of 62- and 8-kDa, respectively, that remain as part of the mature particle (Fig. S1). The resulting mature particles are 42 nm in diameter, and they are morphologically indistinguishable from the authentic virus[10]. Lowering the pH of a procapsid preparation from 7.6 to 5.0 results in rapid, large-scale, conformational changes of the subunits and maturation[11]; however, intermediate capsid states can be detected at pH values maintained between 7.6 and 5.0[12]. These conformational changes are reversible at elevated pH if no more than 10% of the subunits have been cleaved or in the mutant Asn570Thr, which is incapable of effecting the cleavage[12,13].

The γ peptide acts as a lytic peptide, providing an effective mechanism to breach a membrane and allowing non-enveloped animal viruses to enter cells[14]. Experiments using liposomes loaded with a self-quenching dye, and NωV VLPs at different stages in the maturation (pH 7.6–5.0), have shown that the maximum lytic activity occurs at alkaline pH (pH 7.5 and above) and that it is necessary for the γ peptide to be released for this activity[15,16]. Therefore, procapsid VLPs, in which the α peptide is not cleaved, do not lyse liposomes[16]. This pH-dependent functionality of the lytic peptide is believed to correlate with the uptake of virus particles under the alkaline conditions of the insect gut[17].

Though comparison of the structures of procapsids and capsids produced by expression in insect cells and subsequent in vitro maturation have been very revealing, they have led to only an understanding of the gross structural changes that accompany the process. Furthermore, they rely on the assumption that maturation by reducing pH follows an identical pathway to that which occurs within cells. While this is reasonable, given that structures of capsids matured in vitro are indistinguishable from authentic virions isolated from infected insects, it is possible that the pathway may be different within cells. However, maturation of capsids within insect cells has not been observed, rendering in vivo studies impossible in this system.

Plants are a highly effective system for producing VLPs[18–23]. In most instances, expression of the coat protein subunits alone results in the assembly of VLPs; however, in the case of members of the order *Picornavirales*, where coat protein cleavage is required for particle assembly, co-expression of a virus-encoded protease is also required[20,22]. To date, the resulting purified VLPs have been mature, static structures, which closely resemble the original virion both in structure and immunogenicity.

To explore whether plant-based systems can be used to visualize the molecular details of the maturation processes, we transiently expressed the coat protein subunit of NωV and show that the cleavage associated with maturation occurs within the cells. It proved possible to purify both procapsids and capsids from leaf tissue by varying the time at which the VLPs were extracted. Using cryo-EM, we solved the structure of the procapsids and capsids to 6.6 and 2.7 Å resolution, respectively, and showed that the plant-produced mature capsids have the ability to lyse liposomes and are essentially identical in structure to authentic virus. Comparison of the structures of the plant-produced procapsids and capsids revealed that, in addition to large scale rigid body motions, internal regions of the subunits are extensively remodelled during maturation, creating the active site required for autocatalytic cleavage and infectivity.

## Results

**Expression and purification of NωV VLPs in plants.** The NωV WT coat protein gene, codon-optimised for *N. benthamiana*, was inserted into plasmid pEAQ-*HT*[24] to give plasmid pEAQ-*HT*-NωV-WT. Expression of the coat protein was initiated by infiltrating leaves of *N. benthamiana* with suspensions of *A. tumefaciens* harbouring pEAQ-*HT*-NωV-WT followed by the collection of leaf disks various days post-infiltration (dpi). Western blot analysis of total protein extracts using a polyclonal antibody specific for the NωV coat protein showed that expression of the 70 kDa full-length coat protein (α protein) was first detected at 2 dpi (Fig. 1a). By 3–4 dpi a lower molecular weight band of about 62 kDa, consistent with the size of the β protein, appeared suggesting that the full-length NωV coat protein undergoes processing over time within the plant tissue. Additionally, higher molecular weight bands of around 150 kDa were observed in the extracts from later time points, suggestive of the formation of dimers, whether covalently linked or just SDS-resistant, of the coat protein. The identity of these bands as consisting of NωV coat protein was confirmed by mass spectrometry of tryptic digests of the purified proteins.

To confirm that VLPs are, indeed, produced in plant cells, thin sections were prepared from leaves 8 days after infiltration with either pEAQ-*HT* (empty vector) or pEAQ-*HT*-NωV-WT and examined by TEM. Large numbers of particles, characteristic of NωV VLPs, could be seen in the cytoplasm of cells from leaves infiltrated with pEAQ-*HT*-NωV-WT (Fig. 1c, d) but not in cells from tissue infiltrated with the empty vector (Fig. 1b). This, coupled with the time-dependent cleavage of the α protein, indicates that the expressed NωV coat protein subunits assemble into procapsids that are able to mature within plant cells.

To characterise the particles produced in plants, VLPs were extracted at pH 7.6 either 3–4 dpi to maximise the level of procapsids[25] or at 8 dpi to isolate mature capsids. Both types of capsid particle were purified by centrifugation through continuous gradients (10–40% (w/v) sucrose for procapsids, and 10–50% (w/v) Optiprep for mature capsids), and the protein content of the fractions was analysed by SDS-PAGE (Fig. S2). Fractions containing the uncleaved α protein and the cleaved β protein were pooled separately to give samples of purified procapsids and capsids, respectively. Visualisation by negative stain TEM showed that the purified procapsids had a diameter of ~48 nm and appeared porous due to penetration of stain within their cores and were heterogenous (Fig. 1e), while the mature capsids were compact and impermeable to stain, with a diameter of ~42 nm (Fig. 1f). These morphologies are similar to those observed for the equivalent NωV VLPs produced in insect cells[13]. For both procapsids and mature capsids, the yield of VLPs was in

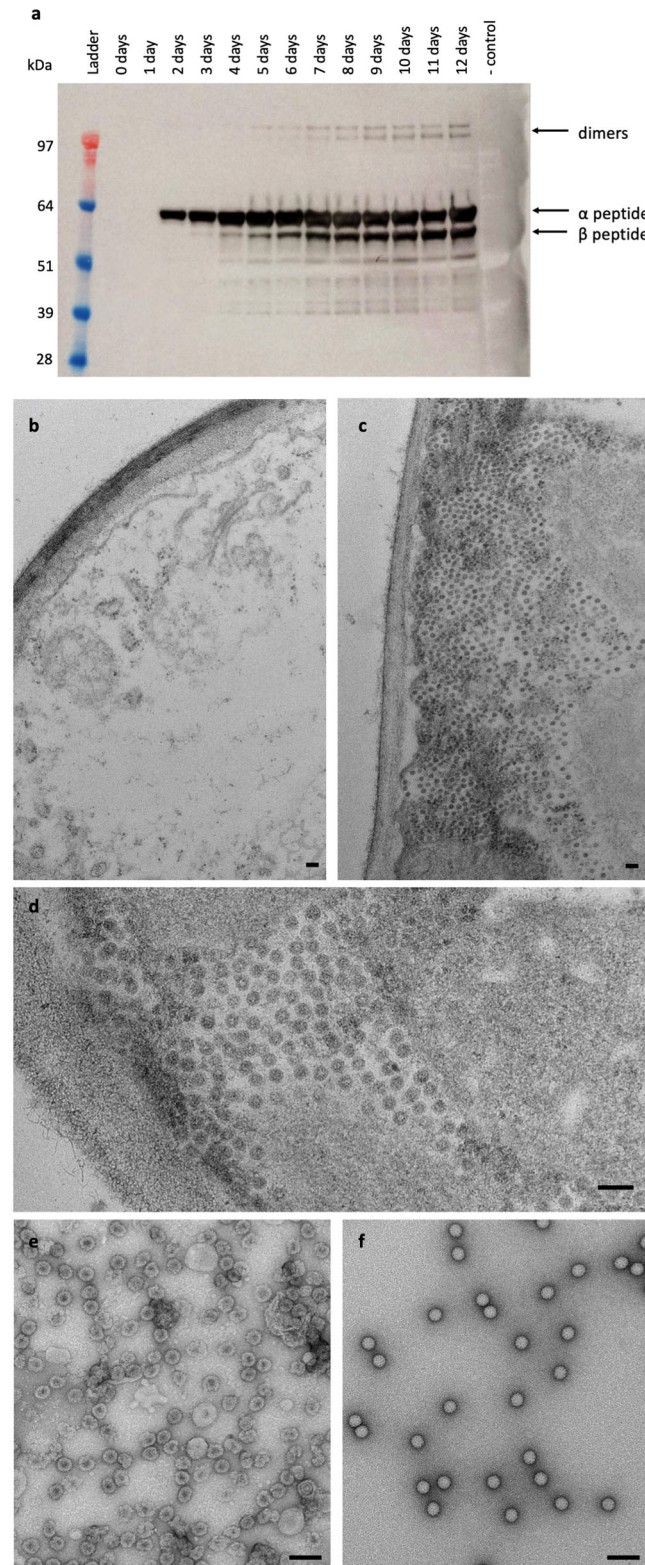

**Fig. 1 Expression of NωV coat protein in *N. benthamiana*. a** Western blot of samples collected 0–12 dpi from leaves infiltrated with pEAQ-*HT*-NωV-WT. The negative control (- control) was from leaf material infiltrated with pEAQ-*HT* (empty vector). The protein was detected using a polyclonal antibody for the NωV coat protein. The positions of the uncleaved (α) and cleaved (β) versions of the coat protein are indicated, as are the positions of the dimeric forms. Ladder = SeeBlue Plus 2 pre-stained protein standards. Electron micrographs of thin sections of leaves 3 dpi with pEAQ-*HT* (**b**) or pEAQ-*HT*-NωV-WT at two different magnifications (**c**, **d**). All the leaf sections were counter-stained with 2% (w/v) uranyl acetate and 1% (w/v) lead citrate. Electron micrographs of purified procapsids (**e**) and capsids (**f**) negatively stained with 2% (w/v) uranyl acetate. For all the micrographs the scale bar = 100 nm.

SDS-PAGE (Fig. 2a). This showed that cleavage occurred with a half-life of 45–60 min, compared with a half-life of 30 min for a sample of procapsids produced in insect cells that was analysed in parallel (Fig. 2b), the latter result being consistent with that reported previously[26]. Coat protein dimers also shifted in size, indicating that the dimerisation does not inhibit cleavage (Fig. 2a). As autocatalysis depends on the precise positioning of amino acids to a fraction of an Ångstrom[27], this finding demonstrates the fidelity of NωV VLP assembly in plants.

To assess the functionality of the γ peptide released during the pH-induced maturation of plant-produced procapsids in vitro, samples were mixed with DOPC liposome preparations at pH values from 5.0 to 9.0. As a control, VLPs produced in insect cells matured at pH 5.0 were analysed in parallel. These membrane disruption assays demonstrated that mature VLPs from both plants and insect cells have a similar lytic activity at alkaline pH (Fig. 3a). However, the plant-produced NωV VLPs had a slightly higher initial rate of liposome lysis than the insect-cell produced particles (Fig. 3b).

**Structure analysis of NωV VLPs**. NωV procapsids purified from plants were analysed by cryo-electron microscopy. The particles were structurally heterogeneous, with many being broken or distorted (Fig. 1e and S3a), thereby limiting the resolution of the subsequent reconstruction. However, at 6.6 Å, the plant-produced procapsid model is the highest resolution structure of the NωV procapsid currently available (Fig. 4a, c); the previously published NωV procapsid structure being at only 28 Å resolution[10].

The more robust capsid VLPs (Figs. 2f and S3b) enabled a 3D reconstruction at 2.7 Å resolution, the highest resolution structure available for NωV (Fig. 4b, d). Comparison with the 2.8 Å resolution crystal structure from authentic virions (PDB 1OHF) revealed that the virion and plant-expressed VLP capsids are virtually indistinguishable, with an rmsd in Cα atoms of 0.461 Å between asymmetric units (ASU; based on 2256 aligned residues). Thus, this capsid structure demonstrates that NωV VLPs produced in plants must initially assemble in a way that facilitates the authentic maturation pathway, and that plant-based expression systems are therefore able to support the complex maturation pathways of animal viruses.

Figure 5 shows the density of the procapsid (a) and capsid (b) rendered in colours that depict the 4 subunits in the icosahedral asymmetric unit; A (blue), clustered around the icosahedral five-fold axes and B (red), C (green) and D (yellow) clustered around the quasi-3-fold axes. Figure 5c, d show the change in density (viewed from inside the particle) between procapsid (c) and capsid (d) at the quasi-2-fold axes that relate an A, B, C trimer to the D, D, D trimer.

The model derived from the procapsid density was initially based on the rigid body fit of the jellyroll and Ig domains (residues 117–532) into the envelope of density that clearly

the range 0.1–0.25 mg of purified protein per gram of fresh weight infiltrated leaf tissue.

**Functional properties of plant-produced NωV VLPs**. To determine whether the α protein in plant-produced procapsids can undergo the autocatalytic cleavage associated with maturation in vitro, the pH of a suspension was rapidly reduced from pH 7.6 to pH 5.0 and the extent of α protein cleavage monitored by

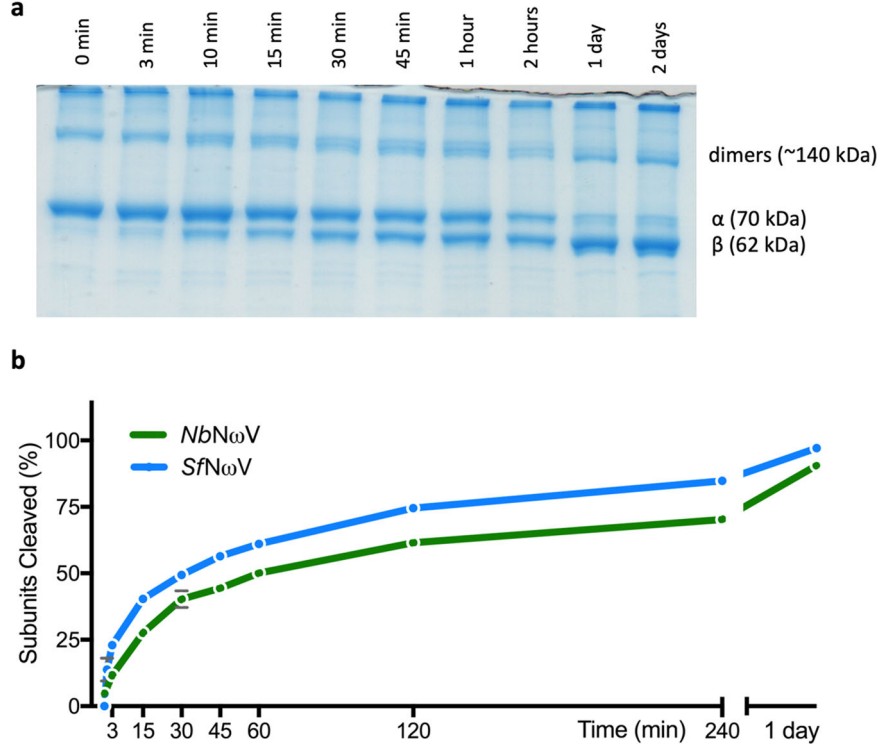

**Fig. 2 Autocatalytic cleavage analysis. a** SDS-PAGE analysis of plant-produced procapsids incubated for different times at pH 5.0. The positions of the uncleaved α protein and the autoprocessed β protein are indicated as are the dimeric forms. **b** Comparison of the kinetics of cleavage of NωV procapsids from plants (green line) and insect cells (blue line). The fraction of subunits cleaved was determined by band densitometry of the stained gel image using the software ImageJ. A square of fixed size was used to delimit the gel area containing α and β proteins. Each time point was measured individually, and the programme retrieved the density of each protein band. The sum of α and β signals corresponded to 100%. Therefore, the extent of cleavage corresponds to the percentage of β density in relation to the total density of each time point. The raw data for the plot is given in Supplementary Data 1.

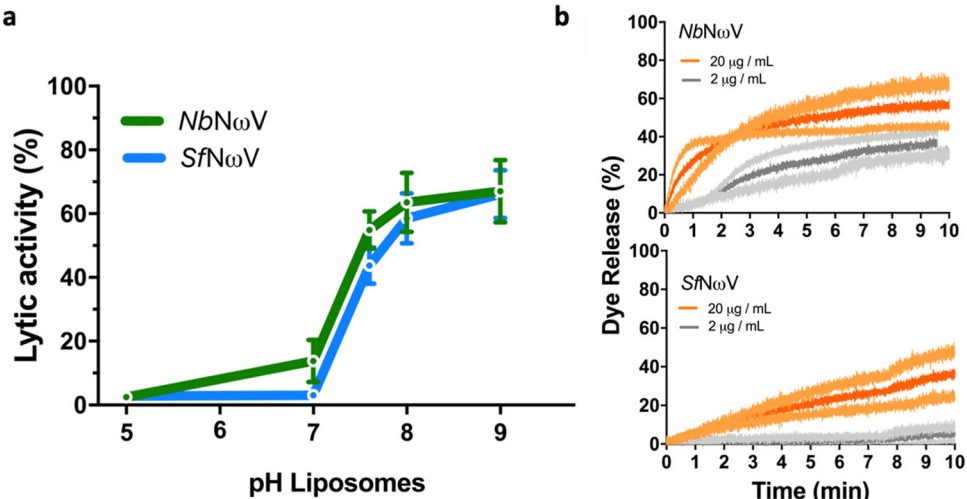

**Fig. 3 Membrane lytic activity of NωV VLPs produced in plants and insect cells. a** Lytic activity of mature NωV VLPs expressed in plants (green line) and insect cells (blue line) against DOPC liposomes under different pH conditions (from 5.0 to 9.0). End point values were measured after 10-min incubations with liposome preparations. Standard deviations from at least two experiments are shown as error bars. The raw data for the construction of the curves is given in Supplementary Data 2. **b** Initial kinetics of liposome dye release at pH 7.6 induced by mature NωV VLPs produced in plants (top panel) or insect cells (bottom panel). In each case, two NωV VLP concentrations were tested, 2 µg/mL (grey lines) and 20 µg/mL (orange lines). Standard deviations from at least two experiments are shown as lighter shades of each colour.

defined their positions and orientations in all four subunits within the icosahedral asymmetric unit. The N-terminal residues (44–116) and C-terminal residues (533–644), poorly defined in the previous procapsid reconstruction[9], were modelled with approximate reference to the equivalent residues clearly defined

in the capsid. The modelling was aided by the equivalence of the density in the terminal regions in all four subunits of the procapsid, which is not the case in the capsid (Fig. 6). Treating the four subunit models as equivalent reduced the parameters and improved the confidence in the final coordinates.

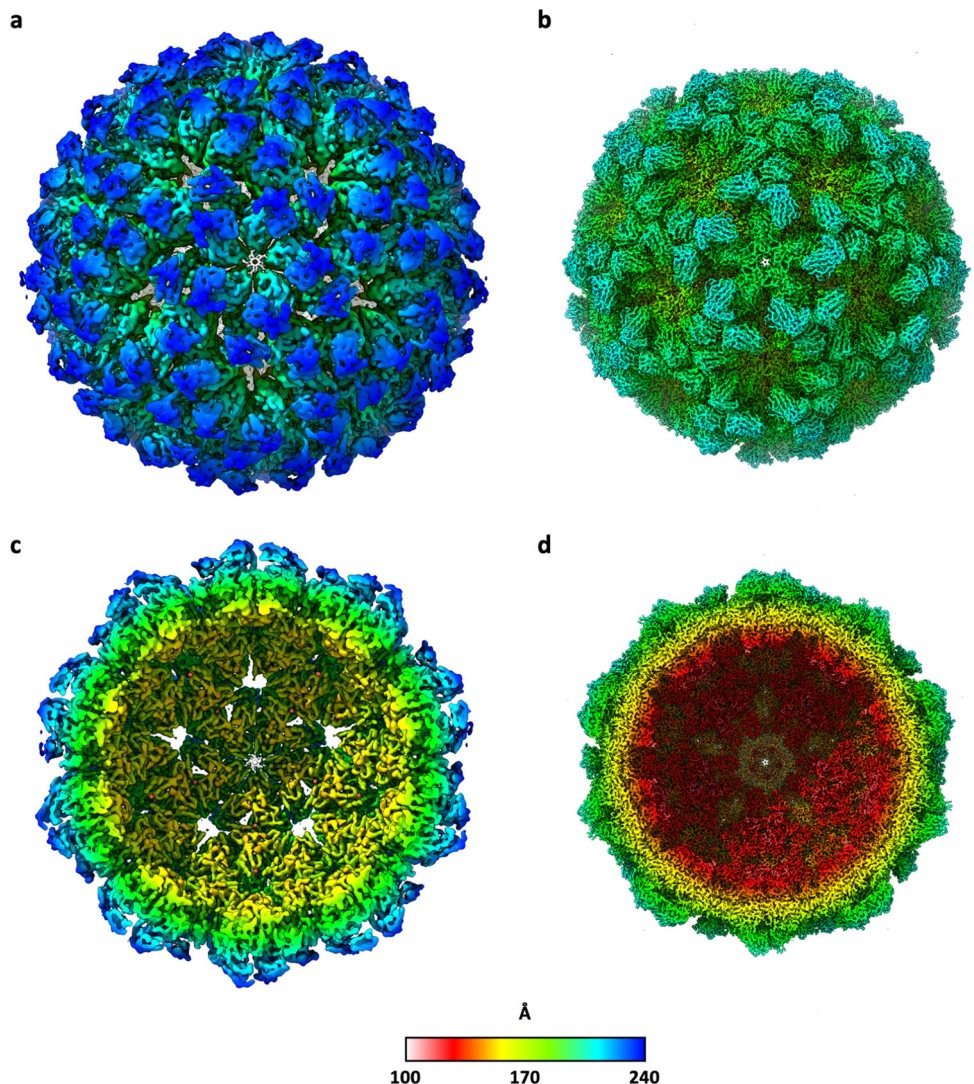

**Fig. 4 Radially coloured cryo-EM reconstructions of NωV VLPs highlighting the degree of compaction on transitioning from the procapsid to the capsid.** Full and cutaway views of the procapsid reconstruction at 6.6 Å resolution (**a**, **c**, respectively), and for the capsid at 2.7 Å resolution (**b**, **d**, respectively) are shown as viewed down the icosahedral five-fold axis. The surfaces are coloured with respect to the distance from the centre of each particle according to the key below.

## Discussion

The present study demonstrates that it is possible to recapitulate the complex maturation process involved in the production of NωV VLPs in plants. While plants have previously been used to make capsids of viruses which require processing of the precursor protein by a viral protease, such as cowpea mosaic virus and poliovirus[20,22], the production of a VLP that can undergo large-scale structural rearrangements and effect a precise autoproteolytic cleavage has not previously been demonstrated in plants, and so the results described here for NωV are unique.

When the NωV coat protein was expressed in plants, cleavage of the α protein increased with the time that the particles remained in the plant. These results were unexpected since insect cell expression results exclusively in the production of procapsids, containing the uncleaved α protein, when VLPs are purified at pH 7.6[9,10], with cleavage only occurring upon acidification in vitro. The autocatalytic cleavage of non-enveloped virus coat proteins has usually evolved to occur in the extracellular environment, avoiding the generation of membrane-disruptive peptides inside the already-infected host cell[28]. The *in planta* maturation observed here could arise from acidification of the environment over time. Although the pH of the plant cytoplasm is believed to be around 7.5, other compartments, such as the apoplast and vacuole, are more acidic[29,30]. Thus, tissue senescence could result in acidification of the environment where the VLPs accumulate. However, alternative explanations, such as an altered concentration of RNA in plant-expressed VLPs triggering maturation cannot be ruled out, as such a role for RNA has been found in the closely related tetravirus, *Helicoverpa armigera* stunt virus (HasV)[31]. Insect cell-expressed VLPs have been shown to contain host-derived RNA[32] and our preliminary results indicate that the plant-made VLPs also contain RNA, though this has not been characterised in detail.

Although the in vitro maturation and liposome lysis experiments demonstrate the biological functionality of plant-produced NωV VLPs, some differences in the kinetics of both processes are apparent. The rate and efficiency of autocatalytic cleavage were both lower than found with those made in insect cells. By contrast, plant-derived particles have slightly higher initial lytic activity than insect-produced VLPs. These differences could be a consequence of differences in the dynamic properties of the particles produced in the two systems. One plausible cause of

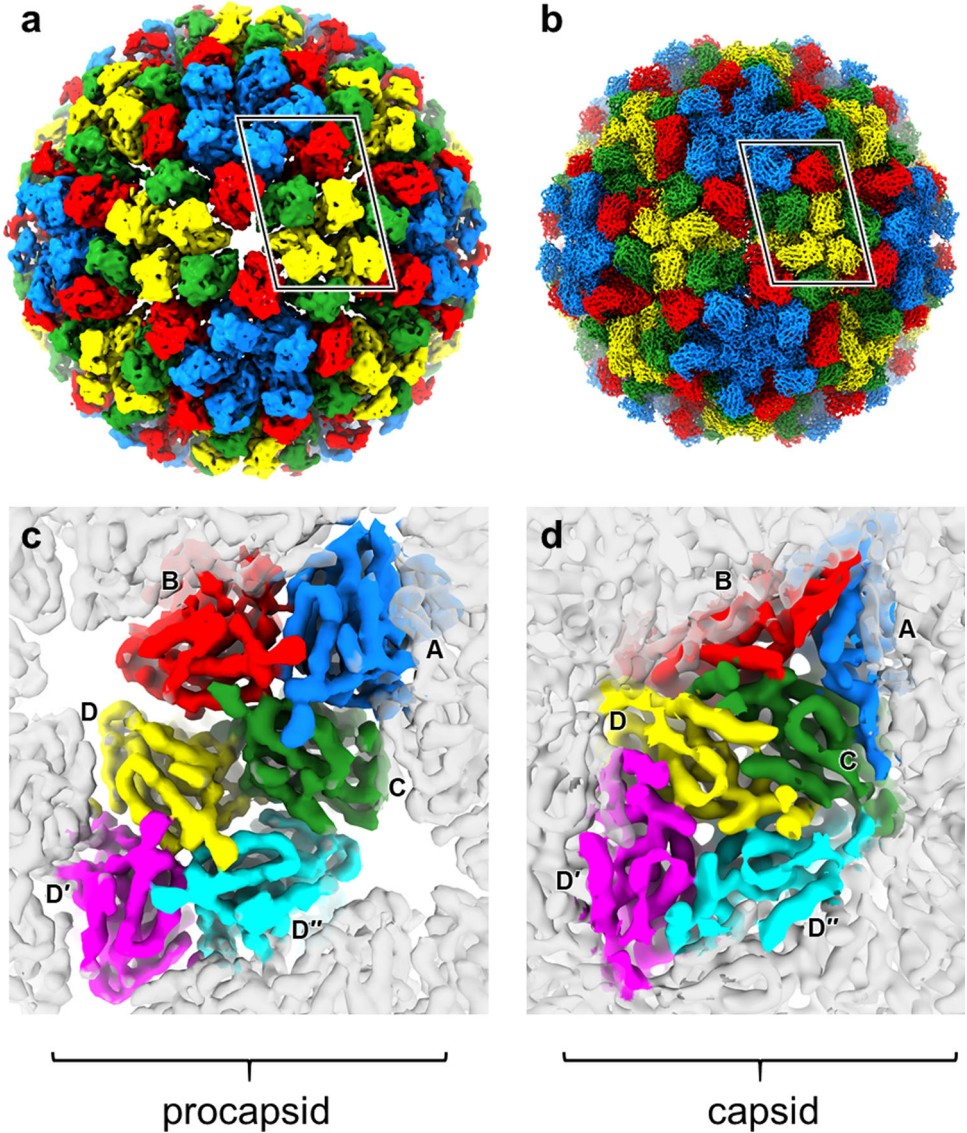

**Fig. 5 Cryo-EM reconstructions of NωV VLPs highlighting the quaternary structures.** Reconstructions of **a** the procapsid at 6.6 Å resolution and **b** the capsid at 2.7 Å resolution as viewed down the icosahedral two-fold axis. The density associated with four 70 kDa subunits that comprise the icosahedral asymmetric unit are coloured: A—blue, B—red, C—green and D—yellow. The highlighted region is shown below as viewed from the core of the particle for **c** the procapsid and **d** the capsid. The density for the rest of the particle is shown in semi-transparent grey. For ease of comparison the capsid density has been down-sampled to 6.6 Å resolution in **d**. In both the lower panels, subunits A–C are arranged around a quasi-three-fold axis and three copies of subunit D are arranged around the icosahedral three-fold axis (D–yellow; D′–magenta; D″–cyan; N.B. these are all yellow in **a**, **b**—different colours are used here to clearly define the densities associated with each subunit). Since icosahedral symmetry was imposed during reconstruction, the density associated with the three D-type subunits will be identical within each map. It is clear from these comparisons that the symmetry is very strong around the quasi-three-fold axis (A–C) in the procapsid but is broken in the capsid. Indeed, the subunit conformations and arrangements around the two types of three-fold axes are closely similar for the procapsid (A–C vs. D–D″). Additionally, these views highlight the porous packing seen in the procapsid relative to the highly compact arrangement of subunits in the capsid.

such changes could be the coat protein dimers that are present only in plant-derived particles; these could affect the efficiency of the quaternary rearrangements which accompany maturation. The alternative explanation, that the plant-produced samples contain a contaminant capable of lysing liposomes, is less likely as no lysis occurred when procapsids produced in plants were used in the assay.

We were able solve the structure of plant-produced procapsids to 6.6 Å resolution and the structure of mature capsids to 2.7 Å resolution allowing comparison of the structures before and after maturation. The most dramatic aspect of the transition from procapsid to capsid is the ~30 Å translation-rotation of the

subunits from a maximum radius of 240 Å to the capsid radius of 210 Å. Due to the 6.6 Å resolution of the procapsid density and the remarkable correspondence when the density of the four subunit volumes was superimposed, in the refinement procedure structural equivalence was enforced on the four procapsid subunit models in the icosahedral asymmetric unit. The 2.7 Å resolution of the capsid allowed subunit models to be constructed and refined independently with no symmetry constraints within the icosahedral asymmetric unit. Figure 7 depicts the large-scale transitions graphically, with selected amino acids lying close to the geometric centres of the Ig, jellyroll and helical domains of the procapsid and capsid states and the vectors connecting these

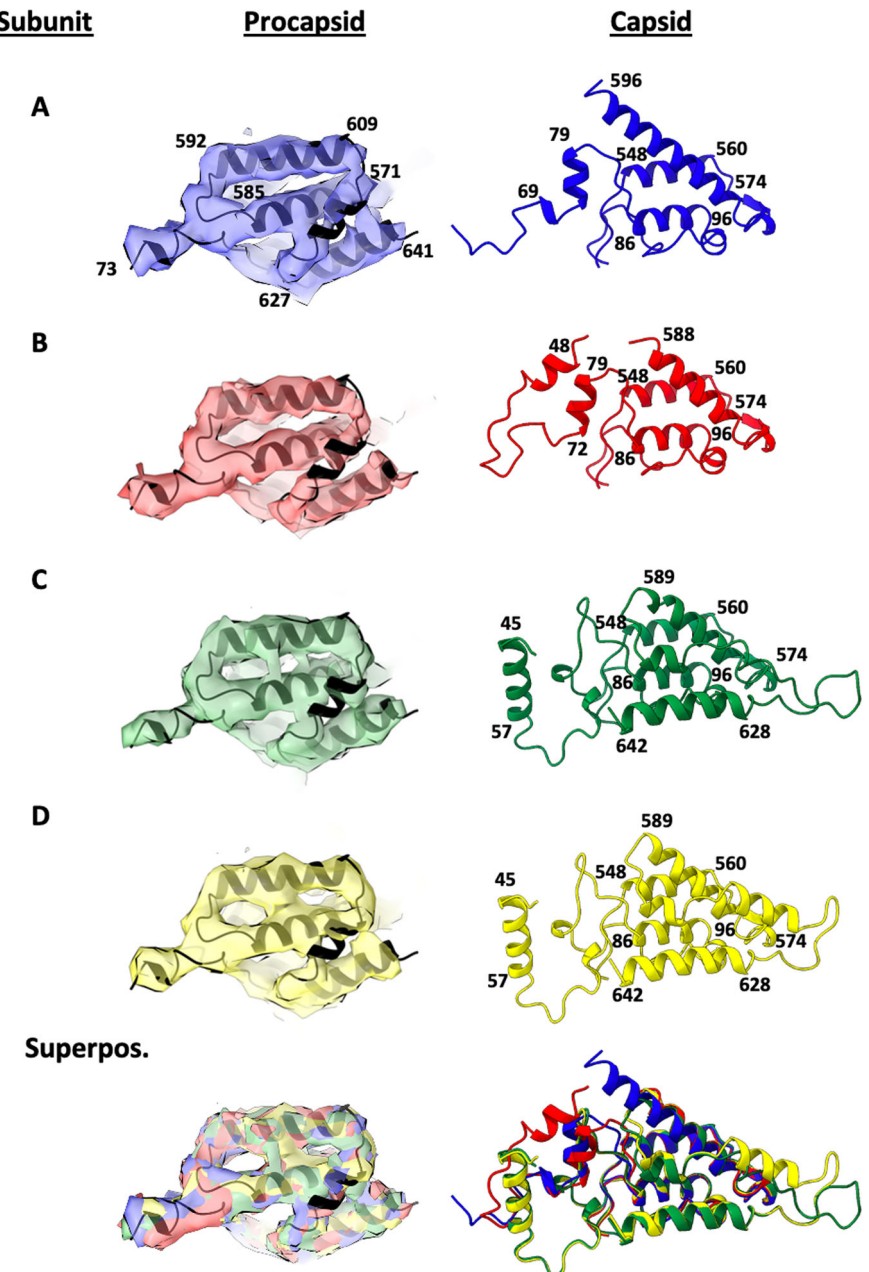

**Fig. 6 Internal helical domains of procapsid and capsid subunits in one icosahedral asymmetric unit.** Left: Procapsid cryo-EM density (6.6 Å resolution) and model for the internal helical domain (N-terminal and gamma peptide) of subunits A, B, C, D all viewed in the same orientation. The model of the internal helical domain (in black), refined as an identical rigid body for all four subunits, precisely fits the unique cryo-EM densities for the four subunits. Superposition of the four densities and models (bottom left) demonstrates the closely similar structures in this region of the procapsid. Right: Refined capsid models (2.7 Å resolution) of the internal helical domain in each of the four subunits and their superposition (bottom) reveals large differences from each other and the procapsid demonstrating substantial refolding of these sequences during maturation.

equivalent residues, being represented. Upon first inspection, the variations in vector lengths for the different subunit pairs (Fig. 7b, c) suggest that the domains move relative to one another and along different trajectories depending on the subunit type. However, a superposition of the four capsid subunit types onto the procapsid subunit reveals that equivalent marker atoms correspond closely in position. This shows that the relative placement of the three domains is essentially static and therefore the differences in the trajectory lengths between the subunit pairs mostly reflect different combinations of rigid body translations and rotations of the subunits rather than conformational rearrangements that are only seen in the helical region.

Regardless of the model interpretation in the procapsid, it is striking that the interior density in regions that must be occupied by the N and C terminal regions is virtually identical in all four subunits in the icosahedral asymmetric unit (Fig. 6). This is not the case in the capsid, demonstrating that subunits transition from being virtually equivalent in the initial procapsid assembly product to being non-equivalent in the capsid. This requires that, in addition to the large rigid body motions described above, the N and C-terminal regions that comprise the inner helical domain refold into different conformations despite having identical amino acid sequences. The refolding has two consequences obvious in the capsid structure. First, the active site for the

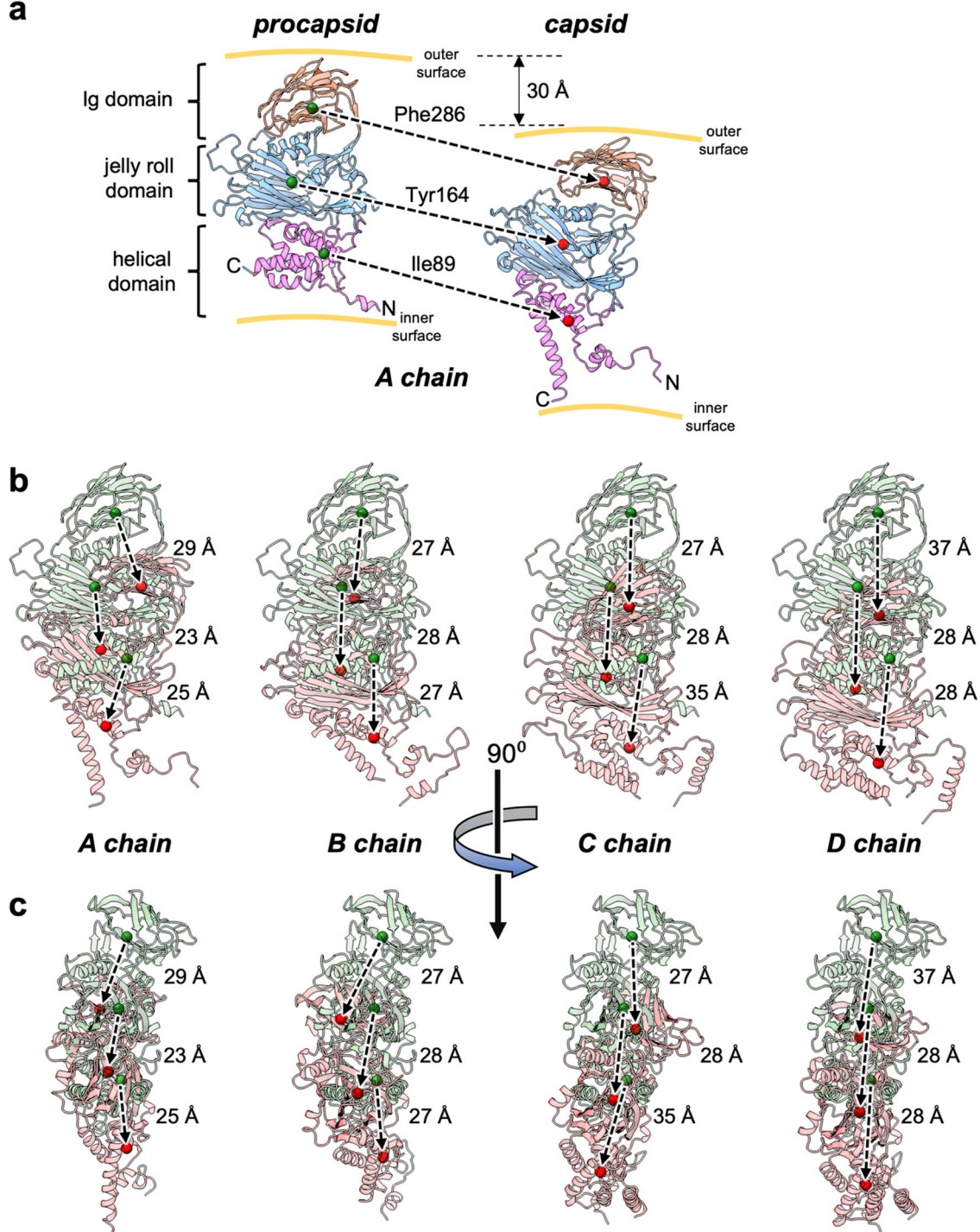

**Fig. 7 Trajectories of subunit motions on maturation from procapsid to capsid.** In addition to the significant refolding of the helical domain during capsid maturation, the NωV subunits are translated through space by ~30 Å as the protein coat contracts from a diameter of roughly 480 to one of 420 Å. To track the domain movements, we have defined three marker atoms that correspond to the Cα atoms of equivalent residues in each subunit that lie close to the geometric centres of the three domains. **a** The protein backbone is shown as a semi-transparent cartoon and the marker atoms are shown as solid spheres (procapsid = green; capsid = red) with the three domains distinguished by different colours. For clarity in (**a**) only, the procapsid and capsid positions (for chain A only) have been separated horizontally, whereas in all the other panels their true relative positions are shown after superposition of the full procapsid and capsid icosahedra. For ease of comparison, in each of (**b**, **c**), the subunit pairs have been oriented with respect to the same procapsid subunit view, with procapsid and capsid cartoons distinguished by green and red colouration, respectively. The trajectories of the marker atoms are indicated by dashed arrows. The large structural rearrangements in the helical domains occur mostly at the periphery where they interface with neighbouring subunits or form the inner surface of the particle; the core of this domain (including the marker atom) is essentially unchanged. Furthermore, superposition of all the capsid subunits onto the procapsid subunit reveals that equivalent marker atoms overlap closely. This indicates that the relative placement of the three domains remains effectively unchanged and therefore the differences in the trajectory lengths between the subunit pairs mostly reflect different combinations of rigid body translations and rotations of the subunits rather than conformational rearrangements.

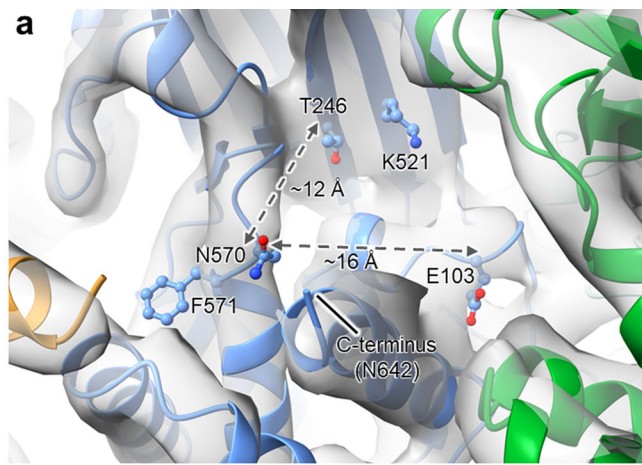

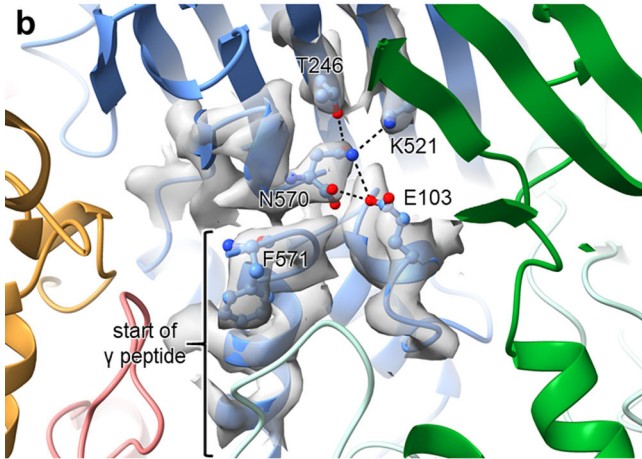

**Fig. 8 Structural remodelling at the NωV autocatalytic cleavage site.**
Close-up views of the cleavage site regions in the **a** procapsid and **b** capsid
models revealing how disparate the key catalytic residues, Asn570 and
Glu103, are in the procapsid as compared to their juxtaposition in the
capsid following cleavage of the γ peptide (important residues are shown in
ball-and-stick representation). For both models, the site in the A subunit is
shown, which is coloured light blue, with neighbouring subunits depicted in
different colours. Also shown are the density maps for the procapsid (6.6 Å
resolution) and for the capsid (2.7 Å resolution), although for the latter,
only the density associated with the highlighted residues is shown for
clarity. Note that due to the low resolution of the procapsid reconstruction,
the modelling of atomic coordinates is only approximate, especially for the
protein side chains.

autocatalytic cleavage is formed in all subunits when the catalytic
residue Glu 103 moves to function as a base for the formation of a
cyclic imide and cleavage between residues Asn570 and Phe571
(Figs. 8a, b and S1). Since all the residues required for the cleavage
are on the same polypeptide chain and cleavage does not occur in
the procapsid, a refolding mechanism is the only way to create the
active site. Second, residues 600–644 form extended helices in
only the C and D subunits where they may function as a switch to
promote the flat contacts between the C and D subunits related
by quasi-2-fold axes and also to stabilise the particle (Fig. 5),
though it is difficult to establish cause and effect from the analysis
of two static structures. Indeed, the majority of the large struc-
tural rearrangements in the helical domains occur at the per-
iphery where they interface with neighbouring subunits or form
the inner surface of the particle. The procapsid structure provides
new insights into the refolding required to achieve maturation
and cleavage. Although much more detailed, the overall

description above supports the account by Canady et al.[10] com-
paring the procapsid structure (based on rigid body fitting of four
capsid subunit models into the procapsid 28 Å cryo-EM density)
and the mature virus structure. They proposed that the basic units
of procapsid assembly are weakly interacting solution dimers that
associate into weakly interacting trimers to form the fragile,
spherical, procapsid structure of 240 virtually identical quasi-
equivalent subunits. The fragility of the procapsid at neutral pH is
due to electrostatic repulsions, allowing maturation to the infec-
tious virion to be primed to occur when the pH is reduced to 5.0.
During maturation, the subunits differentiate into tight trimers
(the dominant morphological feature on the capsid surface) sta-
bilised by the associated maturation cleavage. The newly deter-
mined procapsid structure reported here has allowed the
molecular details of this process to be revealed. The stability of
intermediates at different pH values suggests that the two-state
system described here can be elaborated into a "movie" with the
frames determined by the structures at different pH values.

## Materials and methods

**Plasmids.** The sequence encoding the NωV α coat protein (Genbank MT875167),
was codon optimised for *Nicotiana benthamiana* and synthesised by GeneArt (Life
Technologies) with the flanking restriction sites AgeI and XhoI and a Kozak
consensus sequence (TAACA) at the start of the coding sequence[33]. The gene was
cloned into an AgeI/XhoI- digested pEAQ-*HT* plasmid[24] to produce pEAQ-*HT*-
NωV. *Agrobacterium tumefaciens* LBA4404 were transformed with the construct
by electroporation and bacterial suspension were infiltrated into plants as pre-
viously described[23,25]. For the production of NωV α coat protein in insect cells, a
pFastBac vector harbouring the NωV sequence (Genbank MT875167) was used
with the Bac-to-Bac® Baculovirus Expression System (Invitrogen), as previously
described[16].

**Preparation and visualisation of leaf sections.** Infiltrated leaves were cut into 1
mm² fragments and fixed overnight in a solution of 2.5% (v/v) glutaraldehyde in
0.05 M sodium cacodylate, pH 7.3. Subsequent treatment was carried out as
described by Meshcheriakova and Lomonossoff (2019)[34]. The leaf sections were
counter-stained with 2% (w/v) uranyl acetate and 1% (w/v) lead citrate.

**Small-scale protein extraction.** Tissue agroinfiltrated with pEAQ-*HT*-NωV-WT
was harvested at various days post-infiltration and samples of ~110–130 mg of
fresh weigh material were immediately frozen in liquid nitrogen and then stored at
−80 °C. The frozen leaf disks were ground and the powdered tissue was mixed with
200 µl of extraction buffer (50 mM Tris, 250 mM NaCl, pH 7.6) + 100 µl of 4×
NuPAGE LDS sample buffer (Invitrogen) containing β-mercaptoethanol (3:1 ratio)
and immediately heated to 100 °C for 20 min. The samples were centrifuged at
16,000 × g for 30 min and the supernatant analysed on SDS-PAGE gels.

**Particle purification.** Leaves of *N. benthamiana* agroinfiltrated with pEAQ-*HT*-
NωV were homogenised in two and a half volumes of extraction buffer (50 mM
Tris, 250 mM NaCl, pH 7.6) using a waring blender. The crude extract was filtered
through two layers of Miracloth (Millipore) and then centrifuged at 15,000 × g for
20 min at 11 °C. To purify procapsids, the supernatant was centrifuged through a
30% (w/v) sucrose cushion prepared in the same buffer as described by Peyret[35].
The resuspended pellets were clarified at 12,000 × g for 30 min at 11 °C and the
supernatant was ultracentrifuged through 10–40% (w/v) continuous sucrose gra-
dients. To purify mature capsids the clarified extracts were directly centrifuged
through 10–50% (w/v) Optiprep gradients; 10–40% (w/v) sucrose gradients were
also successfully used to produce mature capsids. In both cases the gradients were
prepared in extraction buffer. Gradient fractions containing either procapsids or
mature capsids were identified by SDS-PAGE, pooled and the samples further
purified and concentrated using centrifugal filters (Amicon®, Merck) with a
molecular weight cut-off (MWCO) of 100 kDa. The concentrated VLPs were stored
in the fridge at 4 °C.

Purification of NωV VLPs produced in insect cells was performed as previously
described[16].

**SDS-PAGE and western blot analysis.** Protein extracts were analysed by elec-
trophoresis on 4–12% (w/v) NuPAGE Bis-Tris gels (Life Technologies). The gels
were either stained with Instant Blue (Expedeon) or the proteins transferred to
nitrocellulose membranes for western blot analysis. Specific proteins were detected
using an NωV polyclonal antibody raised in rabbits followed by detection with a
goat anti-rabbit secondary antibody conjugated to horseradish peroxidase and
developed using the chemiluminescent substrate Immobilon Western (Millipore).

**Table 1 Summary of cryo-EM data collection, processing and analysis.**

| Data set | Procapsid | Capsid |
|---|---|---|
| Data collection and processing | | |
| Microscope | Titan Krios | Titan Krios |
| Detector (mode) | Falcon III (integrating) | Falcon III (integrating) |
| Magnification | 75,000 × | 75,000 × |
| Magnified pixel size (Å) | 1.065 | 1.065 |
| Voltage (kV) | 300 | 300 |
| Total dose ($e^-/Å^2$) | 72.0 | 79.5 |
| Defocus range (μm) | −0.7 to −2.7 | −0.8 to −3.0 |
| Movies collected | 8554 | 2788 |
| Particle images | 5426 | 21,395 |
| Symmetry imposed | Icosahedral (I1) | Icosahedral (I1) |
| FSC threshold | 0.143 | 0.143 |
| Map resolution (Å) | 6.63 | 2.72 |
| Map resolution range (Å) | 5.66–13.33 | 2.64–3.77 |
| Refinement | | |
| Initial model used (PDB code) | 7ANM | 1OHF |
| Map sharpening $B$ factor ($Å^2$) | −452.2 | −125.0 |
| Model composition (ASU) | Chains A, B, C, D | Chains A, B, C, D |
| Non-hydrogen atoms | 17,464 | 17,518 |
| Protein residues | 2288 | 2290 |
| R.m.s. deviations | | |
| Bond lengths (Å) | 0.005 | 0.006 |
| Bond angles (°) | 1.02 | 1.04 |
| Validation | | |
| EMRinger score | N/A[a] | 5.1 |
| Molprobity score | 2.93 | 1.59 |
| Clashscore | 22.45 | 4.06 |
| Poor rotamers (%) | 9.68 | 2.08 |
| Ramachandran plot: | | |
| Favoured (%) | 95.2 | 97.1 |
| Allowed (%) | 4.8 | 2.9 |
| Disallowed (%) | 0.0 | 0.0 |
| Ramachandran Z-score | 1.02 ± 0.02 | 1.36 ± 0.02 |
| Fit to map ($CC_{mask}$) | 0.74 | 0.88 |
| Accession codes | | |
| EMPIAR (dataset) | 10555 | 10560 |
| EMDB (maps) | 11911 | 11830 |
| PDB (model) | 7ATA | 7ANM |

[a]The EMRinger score uses the density around side-chains to evaluate the fit of the protein backbone. At a resolution of 6.6 Å there is no discernible side-chain density and thus the EMRinger score is meaningless.

**Transmission electron microscopy of negatively stained particles.** Grids for negative staining were generated by applying 3 μl of sample (~0.1–1 mg/ml) on to carbon-coated 400 mesh copper grids (EM Resolutions). Prior to applying the sample, grids were glow-discharged for 20 s at 10 mA (Leica EM ACE200). Excess liquid was removed, and the grid was stained with 2% (w/v) uranyl acetate for 30 s. Grids were viewed using a FEI Tecnai G2 20 TWIN or FEI Talos 200 C TEM (FEI UK Ltd) at 200 kV and imaged using either an AMT XR-60 or OneView 4k × 4k digital camera (Gatan).

**Autocatalytic cleavage assays.** One volume of NωV procapsids suspended in 10 mM Tris-HCl, 250 mM NaCl, pH 7.6 was mixed with 9 volumes of 100 mM NaOAc, 250 mM NaCl, pH 5.0. The reactions were incubated at room temperature and stopped by adding protein loading buffer and immediately freezing the mixture with liquid nitrogen. The SDS-PAGE analysis was used to quantify cleavage by densitometry analysis of the stained gels as previously described[26].

**Membrane disruption assays.** Liposomes composed of 1,2-dioleoyl-sn-glycero-3-phosphocholine (DOPC; Avanti Polar Lipids, Inc.) and containing sulforhodamine B (SulfoB; Invitrogen/Molecular Probes), a fluorescent dye, were prepared as previously described[36]. For the membrane disruption assays, the liposome suspensions in 10 mM HEPES buffer (pH 7.0) were diluted 100 × in the assay buffers: 100 mM Tris, 250 mM NaCl (adjusted to pH 7.5–9.5) or 100 mM sodium acetate, 250 mM NaCl (adjusted to pH 5.0–7.0). The initial fluorescence intensity of the liposome suspension was measured with a Cary Eclipse fluorescence spectrophotometer (Varian), with an excitation wavelength of 535 nm and an emission wavelength of 585 nm. When the reading was stable, the NωV procapsids or capsids were added to the liposome suspension to the required final concentration and incubated for 5–20 min at room temperature. During the incubation, the fluorescence intensity variations were recorded. Finally, Triton X-100 was added to the liposome suspension to a final concentration of 0.1% (v/v) to determine 100% dye release. The analysis of the data was performed as previously described[16].

**Cryo-electron microscopy.** Cryo-EM grids were prepared by applying 3 μl of sample (~0.2–0.4 mg/ml) to 400 mesh copper grids with a supporting carbon lacey film (Agar Scientific, UK) held on an automatic plunge freezer (Vitrobot Mk IV). The lacey carbon was coated with an ultra-thin carbon support film, less than 3 nm thick (Agar Scientific, UK). Prior to applying the sample, grids were glow-discharged for 30 s (easiGlow, Ted Pella). The samples were vitrified by flash-freezing in liquid ethane, cooled by liquid nitrogen.

Data were collected on an FEI Titan Krios EM at 300 kV (Astbury Biostructure Laboratory, University of Leeds). The exposures were recorded using the EPU automated acquisition software on a FEI Falcon III direct electron detector. Micrographs were collected at a resolution of 1.065 Å/pixel. Movie stacks were motion-corrected and dose-weighed using MOTIONCOR2[37] (Fig. S3). CTF estimation was performed using GCTF[38] and particles were picked using RELION[39,40]. The autopicking was performed using 2D templates generated after an initial run without reference templates (Laplacian). Subsequent data processing was carried out using the RELION 2.1/3.0 pipeline[39–41] (Figs. S3 and S4) with the imposition of icosahedral symmetry for the 3D reconstructions. The capsid model was generated first, starting from the previously published crystal structure of the authentic virus (PDB entry 1OHF)[42,43]. The asymmetric unit (ASU), comprised of four protein chains, was rigid body fitted to the sharpened map in Chimera[44]. To expedite computation, for the subsequent steps, the ASU was visualised, manipulated and refined in the context of its eight nearest symmetry copies, denoted ASU8 (Fig. S5), and maps were cropped to cover just ASU8 with a 15 Å border using phenix.map_box in PHENIX[45]. The model was edited using COOT[46] with reference to unsharpened and sharpened maps and refined using phenix.real_space_refine in PHENIX[45] against the latter. An updated ASU8 was generated from the central ASU after each refinement job. Validation of the final model was performed on the full capsid using Molprobity[47] and EMRinger[48] through the PHENIX interface[49]. The final capsid model was used as the starting point for generating the model of the procapsid using a similar protocol. After sharpening in RELION, the resolution of the procapsid reconstruction was estimated to be 7.1 Å. However, after density modification with phenix.resolve_cryo_em in PHENIX[50] this increased to 6.6 Å, giving a concomitant improvement in the map quality, especially in regions where the structure was less well defined. Nevertheless, at this resolution, there was no discernible difference between the density for the subunits, thus non-crystallographic symmetry constraints were imposed to facilitate refinement, which was performed against the density improved map; Ramachandran restraints were also used. A summary of data collection, processing and analysis is given in Table 1 and rmsd values for pairwise superpositions of subunits from the NωV capsid and procapsid structures are shown in Table S1. Comparisons of cryo-EM densities around the quasi-three-fold and icosahedral three-fold axes of capsid and procapsid reconstructions are shown in Fig. S6. Structural figures were prepared using Chimera[44] and ChimeraX[51].

**Reporting summary.** Further information on research design is available in the Nature Research Reporting Summary linked to this article.

## Data availability

The sequence of the NωV α coat protein used in the study is available from Genbank Accession no. MT875167. All the data supporting the cryo-EM structural work have been deposited in the appropriate databases. Specifically, the raw data are available from EMPIAR (accessions 10555 and 10560), the reconstructions are available from EMDB (accessions 11911 and 11830), and the model coordinates are available from the PDB (accessions 7ATA and 7ANM). All data are freely available from the authors.

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

## Acknowledgements

The authors thank Elaine Barclay and Kim Findlay for TEM training and assistance with TEM sample preparation, Maite Vaslin de Freitas Silva for supplying plants and reagents in Brazil and Tsutomu Matsui for discussions on the biophysics of NωV. We thank Matt Byrne and Tom Dendooven for helpful discussions about cryo-EM data processing. At the John Innes Centre, this work was supported by the United Kingdom Biotechnology and Biological Sciences Research Council (BBSRC) Synthetic Biology Research Center "OpenPlant" award (BB/L014130/1), the Institute Strategic Programme Grant "Molecules from Nature—Enhanced Research Capacity" (BBS/E/J/000PR9794), a capital grant award (BBSRC) to establish Cryo-EM capability at the John Innes Centre and the John Innes Foundation. The experiments done at Universidade Federal do Rio de Janeiro were supported by Conselho Nacional de desenvolvimento científico e tecnológico, CNPq; Fundação de Amparo à Pesquisa do Estado do Rio de Janeiro, FAPERJ. The Thermo-Fisher Titan Krios microscopes were funded by the University of Leeds (UoL ABSL award) and Wellcome Trust (108466/Z/15/Z).

## Author contributions

R.C.-G, J.R.S.R., E.L.H., C.A.S. and D.M.L. carried out the experiments and analysed data. G.P.L., J.E.J., T.D. and N.A.R. conceived and directed the project. All authors contributed to the writing and editing of the manuscript.

## Competing interests

G.P.L. declares that he is a named inventor on granted patent WO 29087391 A1 which describes the HyperTrans expression system and associated pEAQ vectors used in this manuscript. All other authors declare no competing interests.
