## [Peer Review File · Communications Biology]

Reviewers' comments:

Reviewer #1 (Remarks to the Author):

In this article, "Plant-expressed virus-like particles reveal the intricate maturation process of a eukaryotic virus", Castells-Graells et al describe the cryo-EM derived structure of procapsids and capsid of *Nudaurelia capensis* omega virus (N ω V) at 6.6 Å and 2.7 Å, respectively. When N ω V coat protein is expressed in insect cells, it assembles into 48nm VLPs that are fragile and porous. This procapsid can undergo an autocatalytic cleavage in its capsid protein when exposed to an acidic pH in vitro, resulting in a mature capsid that is 42nm in diameter. Previously, 28 Å cryo-EM structures of procapsid had been determined by Canady revealing large quaternary and tertiary structure rearrangements in the procapsid required for maturation. Likewise, the mature capsid structure has been determined at 2.8 Å.

In this paper, the authors have used plant-based expression system to study maturation of N ω V. Interestingly, they find that N ω V expressed in *N. Bentha* formed procapsids that underwent processing over time to form mature capsid within the plant cells. Procapsid particles purified from infiltrated leaves also exhibited autoproteolysis in vitro, establishing success with plant-based system to study capsid maturation. Cryo-EM structures were determined for procapsid and capsid particles. The procapsid structure at 6.6 Å is a substantial improvement over the 28 Å procapsid in the Canady et al 2000 JMB paper. The atomic details are more than incremental. In particular, the 30 Å transition in the diameter between procapsid and mature capsid was previously attributed to subunit (A, B) rotations; authors here are able to track the same subunit motion by tracing trajectories of marker atoms on the subunits. In addition to the subunit rotation, it was also proposed that the internal domain (comprised of the N- and C-termini) in the procapsid would need to undergo significant rearrangement in order to condense into smaller mature capsid. With this structure, authors do find evidence that such refolding in the procapsid internal domain would be eminent for maturation.

Minor comments

Line 37 the abstract should define some of the major observations, not tease with "revealed exciting new insights"

Line 112 initiated instead of monitored

Line 276 Do the C and D subunit helices act as a switch for flattening the contacts between subunits or do the form in response to the new geometry. The authors need to clarify correlation and causation.

Line 680 instead of "frequently little to no density" the authors should note that 6.6 Å density is inadequate to model side chain positions

Figure S1 should provide a figure showing the cleavage mechanism, rather than requiring the reader to look up reference 51.

Reviewer #2 (Remarks to the Author):

The manuscript describes the maturation of virus like particles of *Nudaurelia capensis* Omega virus. The capsid protein of this insect virus has been expressed in plants and the structure of both nascent and mature particles solved using CryoEM.

Maturation of virus particles after assembly is very interesting biological phenomenon and its understanding has many potential applications for example for the development of antiviral drugs. Unfortunately it is also difficult to observe and thus we have scarcity of experimental data from mechanistic studies.

Here the authors used a virus with relatively well studied transition/maturation that is induced in vitro by a change in pH. They have introduced a completely new system to study the maturation – expression in plant leaves. Authors show, interestingly, that the whole maturation process can be replicated in plant cells in vivo. Maturation of complex VLPs that require large structural rearrangements and autoproteolytic cleavage has not been shown previously in plants. I find the whole manuscript of high impact and importance to the field of structural virology. The amount of experimental data is adequate and the claims are well supported by experimental data.

I have only few minor technical comments:

- 1] Although it is implied, the manuscript does not specifically mention whether both insect and plant cell produced particles are devoid of entrapped RNA.
 - 2] I find interesting quite dramatic difference in lytic activity between insect and plant produced VLPs. Could some plant derived contaminant also have such lytic activity?
 - 3] Figure 2, 3 – it might help if the blue and green line have legend in the graph, not only in the figure description.
 - 4] line 322 missing p in powdered
 - 5] Why were the nascent VLPs purified using sucrose while the mature VLPs using the Optiprep gradient?
 - 6] I would appreciate more details in the description of negative staining, eg what stain and times were used.
- Other than that I don't have any more comments to the manuscript.

Response to Reviewers' comments:

We thank the reviewers for their careful consideration of the manuscript. Our replies to the various are given in red below, with line numbers relating to those in the revised manuscript.

Reviewer #1 (Remarks to the Author):

In this article, "Plant-expressed virus-like particles reveal the intricate maturation process of a eukaryotic virus", Castells-Graells et al describe the cryo-EM derived structure of procapsids and capsid of Nudaurelia capensis omega virus (N ω V) at 6.6 Å and 2.7 Å, respectively. When N ω V coat protein is expressed in insect cells, it assembles into 48nm VLPs that are fragile and porous. This procapsid can undergo an autocatalytic cleavage in its capsid protein when exposed to an acidic pH in vitro, resulting in a mature capsid that is 42nm in diameter. Previously, 28 Å cryo-EM structures of procapsid had been determined by Canady revealing large quaternary and tertiary structure rearrangements in the procapsid required for maturation. Likewise, the mature capsid structure has been determined at 2.8 Å.

In this paper, the authors have used plant-based expression system to study maturation of N ω V. Interestingly, they find that N ω V expressed in *N. Bentha* formed procapsids that underwent processing over time to form mature capsid within the plant cells. Procapsid particles purified from infiltrated leaves also exhibited autoproteolysis in vitro, establishing success with plant-based system to study capsid maturation. Cryo-EM structures were determined for procapsid and capsid particles. The procapsid structure at 6.6 Å is a substantial improvement over the 28 Å procapsid in the Canady et al 2000 JMB paper. The atomic details are more than incremental. In particular, the 30 Å transition in the diameter between procapsid and mature capsid was previously attributed to subunit (A, B) rotations; authors here are able to track the same subunit motion by tracing trajectories of marker atoms on the subunits. In addition to the subunit rotation, it was also proposed that the internal domain (comprised of the N- and C-termini) in the procapsid would need to undergo significant rearrangement in order to condense into smaller mature capsid. With this structure, authors do find evidence that such refolding in the procapsid internal domain would be eminent for maturation.

We appreciate the Reviewer's positive comments

Minor comments

Line 37 the abstract should define some of the major observations, not tease with "revealed exciting new insights"

This point is well made. We have changed the ending of the abstract to:

Lines 41-43: "has revealed a ~30 Å translation-rotation of the subunits during maturation as well as conformational rearrangements in the N and C-terminal helical regions of each subunit."

Line 112 initiated instead of monitored

This has been changed as suggested and appears at Line 118 on the revised MS.

Line 276 Do the C and D subunit helices act as a switch for flattening the contacts between subunits or do the form in response to the new geometry. The authors need to clarify correlation and causation.

This is actually very difficult to distinguish through a comparison of two static structures presented in the manuscript. We now add a statement to this effect:

Lines 288-291: "...where they may function as a switch to promote the flat contacts between the C and D subunits related by quasi-2-fold axes and also to stabilize the particle (Figure 5), though it is difficult to establish cause and effect from the analysis of two static structures".

Line 680 instead of "frequently little to no density" the authors should note that 6.6 Å density is inadequate to model side chain positions

We agree and now simply say:

Line 706: "the modelling of atomic coordinates is only approximate, especially for the protein side chains."

Figure S1 should provide a figure showing the cleavage mechanism, rather than requiring the reader to look up reference 51.

A new version of Figure S1 has been prepared to show this in Panel (d). We have added the following to the Figure Legend:

Lines 718-720: **(d) Proposed cleavage mechanism⁵² based on known degradation pathways in eye lens proteins⁵⁴. Adapted from Johnson et al. (2021)⁵.**

Reviewer #2 (Remarks to the Author):

The manuscript describes the maturation of virus like particles of Nudaurelia capensis Omega virus. The capsid protein of this insect virus has been expressed in plants and the structure of both nascent and mature particles solved using CryoEM.

Maturation of virus particles after assembly is very interesting biological phenomenon and its understanding has many potential applications for example for the development of antiviral drugs. Unfortunately it is also difficult to observe and thus we have scarcity of experimental data from mechanistic studies.

Here the authors used a virus with relatively well studied transition/maturation that is induced in vitro by a change in pH. They have introduced a completely new system to study the maturation – expression in plant leaves. Authors show, interestingly, that the whole maturation process can be replicated in plant cells in vivo. Maturation of complex VLPs that require large structural rearrangements and autoproteolytic cleavage has not been shown previously in plants. I find the

whole manuscript of high impact and importance to the field of structural virology. The amount of experimental data is adequate and the claims are well supported by experimental data.

We appreciate the comments.

I have only few minor technical comments:

1] Although it is implied, the manuscript does not specifically mention whether both insect and plant cell produced particles are devoid of entrapped RNA.

This is now mentioned specifically, and an additional reference is cited:

Lines 234-237: "Insect cell-expressed VLPs have been shown to contain host-derived RNA³² and our preliminary results indicate that the plant-made VLPs also contain RNA, though this has not been characterized in detail".

Reference ³² now appears on lines 550-552.

2] I find interesting quite dramatic difference in lytic activity between insect and plant produced VLPs. Could some plant derived contaminant also have such lytic activity?

We now discuss this possibility which we feel is unlikely:

Lines 247-249: "The alternative explanation, that the plant-produced samples contain a contaminant capable of lysing liposomes, is less likely as no lysis occurred when procapsids produced in plants were used in the assay".

3] Figure 2, 3 – it might help if the blue and green line have legend in the graph, not only in the figure description.

This has been done and revised versions of the Figures have been incorporated.

4] line 322 missing p in powdered

Thanks – now corrected (Line 335).

5] Why were the nascent VLPs purified using sucrose while the mature VLPs using the Optiprep gradient?

This is basically historic because Optiprep was used first. We now state that sucrose can also be used to isolated mature capsids (lines 351-352).

6] I would appreciate more details in the description of negative staining, eg what stain and times were used.

We were a bit surprised by this comment as there is a whole section in the Materials and Methods entitled "Transmission electron microscopy of negatively stained particles" describing this. However, we now give more details in the legend to Fig.1 (Lines 610-612) in case this is overlooked by the readers and have also proved extra details regarding the staining of the thin sections.

Other than that I don't have any more comments to the manuscript.

As well as responding to the reviewers' comments, we have also taken the opportunity to add a current address for the first author (Lines 21-23) and include an additional reference ⁵ (Lines 482-484) which was published while the current manuscript was in review.

Response to Reviewers' comments:

In their original reviews the reviewers made a number of suggestions which we have previously addressed. I could see no further comments from the reviewers regarding the changes we made and therefore I assume they were satisfactory.